# Seasonal Shifts in the Morphological Traits of Bloom-Forming Cyanobacteria in Lake Chaohu, China

Yangyang Meng [1,2], Min Zhang [1,3,*], Zhen Yang [1,3], Xiaoli Shi [1,3], Yang Yu [1] and Limei Shi [1]

1   State Key Laboratory of Lake Science and Environment, Nanjing Institute of Geography and Limnology,
    Chinese Academy of Sciences, Beijing East Road 73, Nanjing 210008, China;
    mengyangyang21@mails.ucas.ac.cn (Y.M.); zhyang@niglas.ac.cn (Z.Y.); xlshi@niglas.ac.cn (X.S.);
    yyu@niglas.ac.cn (Y.Y.); lmshi@niglas.ac.cn (L.S.)
2   Nanjing School, University of Chinese Academy of Sciences, Tianquan Road 188, Nanjing 100049, China
3   Jiangsu Key Laboratory for Eco-Agriculture Biotechnology around Hongze Lake,
    Jiangsu Collaborative Innovation Center of Regional Modern Agriculture and Environmental Protection,
    Huaiyin Normal University, Huaian 223309, China
*   Correspondence: mzhang@niglas.ac.cn

**Abstract:** Harmful cyanobacterial blooms in eutrophic water bodies pose a major threat to the environment and to human health. The morphological functional traits of cyanobacteria play important roles in maintaining their competitive advantages. To explore the regulatory mechanisms of the morphological functional traits of different bloom-forming cyanobacteria, we performed a one-year-long phytoplankton survey from November 2016 to October 2017 in Lake Chaohu, China. The colony size and cell diameter of the dominant cyanobacteria were measured, and their relationships were analyzed. The results showed that *Dolichospermum flos-aquae* and *Microcystis aeruginosa* were the dominant cyanobacteria in the lake. *Microcystis* was the dominant species during the summer; the growth of *Dolichospermum* growth surpassed that of *Microcystis*, and *Dolichospermum* became the dominant species in the late autumn, winter and spring. From winter to spring, the colony size of *Dolichospermum* decreased from 222.25 μm to 10.51 μm, and the individual cell diameter increased from 6.5 μm to 7.4 μm. From summer to autumn, *Dolichospermum* showed enlarged colony sizes and reduced cell diameters. The *Microcystis* colony size increased from 83.71 μm in the spring to 196.71 μm in the summer and autumn, while cells diameter remained essentially at 3–4 μm from March to October in Lake Chaohu. The relationship between colony size and cell diameter in *Dolichospermum* was significantly positive, while that of *Microcystis* was not significant. These results suggest that *Dolichospermum* may maintain biomass through a trade-off between cell diameter and colony size, and that a flexible morphological regulatory mechanism exists. This study seeks to improve our understanding of how bloom-forming cyanobacteria maintain their dominance by regulating their morphological traits.

**Keywords:** *Dolichospermum*; *Microcystis*; cell diameter; colony size; trade-off

## 1. Introduction

Cyanobacterial blooms are a major global problem that affects water quality and aquatic organisms, as well as animals and humans, by producing cyanotoxins and different malodorous compounds. The formation and maintenance of cyanobacterial blooms are considered to be a synergistic result of their internal physiological and morphological features and external driving forces, i.e., nutrients, temperature, and underwater available light [1]. When the external driving forces change, the regulation of functional traits in these cyanobacteria may play an important role in the maintenance of blooms, e.g., through trade-offs in morphological traits. However, our understanding of the trade-offs in terms of the morphological traits of these bloom-forming cyanobacteria is limited.

In some eutrophic lakes, cyanobacterial biomasses were found to be insensitive or lagged in response to a decrease in nutrient levels [2–4]. Factors related to global changes are usually considered to be the main variables regulating their responses. Moreover, the relative importance of these factors in promoting cyanobacteria is taxon dependent [5,6]. In addition to external environmental factors, the functional traits of bloom-forming cyanobacteria may also contribute to their responses. Functional traits have been shown to be effective factors that drive the response of these cyanobacteria to changes in nutrients or in variables related to global changes [7]. In particular, cyanobacteria can modify their morphological functional traits to adapt to environmental changes and maintain their biomass [8].

In recent years, much effort has been directed toward functional trait-based approaches to community ecology in order to elucidate the ecological mechanisms of phytoplankton community succession and biomass maintenance. Trade-offs between functional characteristics, such as the trade-off between growth rate and cell size of the red tide alga *Alexandrina*, have been shown to play an important role in biomass maintenance [9]. A growing number of studies have focused on the trade-offs between functional traits of phytoplankton. A study in Bodensee Lake found that there were distinct trade-offs between defense and growth speed for phytoplankton, which is called the defense–growth trade-off. Defense and growth rate represent key characteristics of Bodensee phytoplankton; their defense and growth rate of change can effectively prevent competitive exclusion [10]. Li et al. studied the effects of temperature and macronutrients on phytoplankton colonies in three different lakes and analyzed the spatiotemporal trade-off characteristics of temperature, eutrophication and cyanobacterial growth in different lakes during different periods. The study showed that *Microcystis* benefited directly from climate warming and eutrophication, which reveals the mechanisms for spatiotemporal trade-offs [11]. Among bloom-forming cyanobacteria, *Dolichospermum* and *Microcystis* can regulate their morphological characteristics, such as cell diameter and chain length, and their photosynthetic characteristics to maintain their growth and biomass, which indicates that trade-offs among the traits may play important roles in the maintenance of cyanobacterial blooms [8].

Colony formation is often thought of as a passive aggregation of individual cells driven by cell adhesion and as an effective way to reduce predation risk [12,13]. The competitive advantage of cyanobacteria is attributed to their ability to migrate rapidly through the water column by virtue of their buoyancy. In addition, they achieve high resistance to zooplankton predation by forming large colonies [14]. Floating colonies can efficiently harvest light and store nutrients at specific depths [15]. Cell diameter is likely to directly affect colony size [16]. The cell diameter and colony size of cyanobacteria show seasonal changes. The effect of these changes on the maintenance of the cyanobacterial biomass is unclear, and the trade-offs between cell diameter and colony size have not been fully studied [17].

In this study, we hypothesized that bloom-forming cyanobacteria could regulate their individual diameters and colony sizes to maintain their competitive advantages, which could be a mechanism driving the maintenance of cyanobacterial blooms. To explore the spatiotemporal transformation processes of different dominant cyanobacteria and their trade-off mechanisms, we conducted a one-year-long survey in Lake Chaohu to analyze the seasonal and spatial variations in the phytoplankton community and the relationship between individual diameter and colony size.

## 2. Materials and Methods

### 2.1. Study Lake

Lake Chaohu, the fifth-largest freshwater lake in China, is located in the middle of Anhui Province, China ($117°116'46''$–$117°151'54''$ E, $30°143'28''$–$31°125'28''$ N). The surface area of the lake is approximately 750 $km^2$, but this varies according to the water level (maximum depth: 6 m; mean depth: 3 m). Lake Chaohu is in a state of severe eutrophication, and cyanobacterial blooms (primarily consisting of *Microcystis* and *Dolichospermum*) have dominated the lake over the past few decades [18,19].

### 2.2. Sampling Method

To investigate the seasonal shifts in cell diameter and colony size of the dominant bloom-forming cyanobacteria, we performed a monthly investigation from November 2016 to October 2017 in Lake Chaohu, China. The six sites located in the east lake area (sites 1 & 2), central lake area (sites 3 & 4) and west lake area (sites 5 & 6, Figure 1) represent an increasing trend for nutrient levels from east to the west. At each site, integrated water samples were collected by mixing the surface (50 cm below the surface), middle (half of the water depth), and bottom (50 cm above the lake bottom) water using a Ruttner sampler. The environmental parameters (temperature, pH, conductivity and dissolved oxygen) at every sampling site were measured using a multiparameter meter (Model 6600; Yellow Spring Instruments, OH, USA). Total nitrogen (TN) and total phosphorus (TP) were analyzed by peroxydisulfate oxidation, and permanganate index ($COD_{Mn}$) was measured using the most recent method reported in the literature [20]. Transparency (SD) was measured with a Secchi disk. A total of 72 Phytoplankton samples (500 mL) were collected and fixed with an acid Lugol solution. After being left undisturbed for 48 h, the supernatant was removed and concentrated to a volume of approximately 30 mL, and the phytoplankton were identified and counted under a 40 x-magnification microscope (Imager A2 x, ZEISS, Oberkochen, Germany) [21]. Identification and counts were performed at the species or genus level using the most recent literature and updated information from the AlgaeBase website [22,23]. The biovolume was calculated from the measurements of 30 organisms of each species at each site [24]. The biomass was determined as the algal volume for each site and converted to fresh weight, assuming a specific gravity of 1 g/cm$^3$ [25].

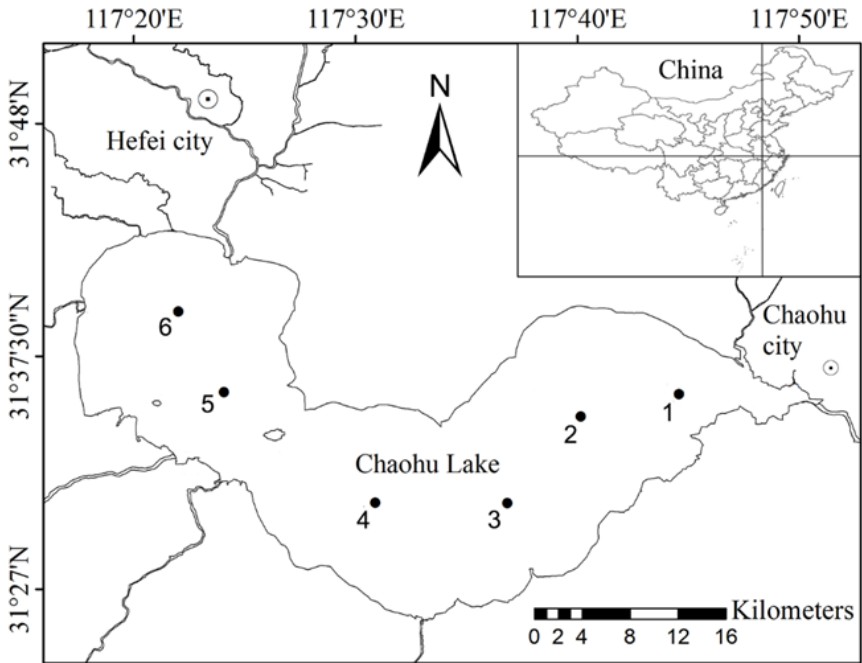

**Figure 1.** Map of Lake Chaohu showing the locations of six sampling sites.

The cell diameters of *M. aeruginosa* and *D. flos-aquae*, the two dominant cyanobacterial bloom-forming species, were measured and analyzed by collecting at least 50 individuals from each sample. Photomicrographs of the samples were taken using a ZEISS AxioCam HRc digital camera (ZEISS, Oberkochen, Germany) coupled with a microscope at ×100 and ×400 magnification and were analyzed using Image-Pro Plus 6.0 software (Media Cybernetics, Inc., Rockville, USA). Colony size was measured with a laser particle size analyzer (Mastersizer 2000, Malvern, UK).

### 2.3. Statistical Analysis

Statistical analysis and related graphs were performed on Origin 2017 and the R language platform R. 4.2.0 [26]. The differences in the cell diameters for *Dolichospermum* and *Microcystis* over the various months were determined using one-way analysis of variance (ANOVA). Differences were considered significant at $p < 0.05$. The colony sizes for *Dolichospermum* and *Microcystis* were extracted by combining the results for colony size and phytoplankton composition. The relationships between cell diameters and colony sizes in *Dolichospermum* and *Microcystis* were assessed by linear regression and a generalized linear model. Before analyzing the corresponding relationships between cell diameter and colony size, the colony size of each taxon was log-transformed to minimize differences in scale and to achieve normality and variance homogeneity.

## 3. Results

### 3.1. Monthly Variations in Environmental Factors

The means and standard deviations of environmental parameters and nutrients from monthly water samples collected at three regions are shown in Table 1. The spatial and temporal distribution characteristics of TN, TP, $COD_{Mn}$ and water temperature are presented in Figure 2. In space, the nutrients showed decreasing trends from west to east. These parameters also showed obvious seasonal variations. TN concentration was higher in winter than those in other seasons. TP concentration and $COD_{Mn}$ usually were lower in spring than those in other seasons.

**Table 1.** Environmental parameters summarized as the mean values $\pm$ standard error in Lake Chaohu from November 2016 to October 2017. The letters indicated the significance of the differences among the lake regions. TN: Total nitrogen, TP: total phosphorus, $COD_{Mn}$: permanganate index, SD: Transparency and Depth: water depth.

|  | TN (mg/L) | TP (mg/L) | $COD_{Mn}$ (mg/L) | pH | SD (cm) | Depth (m) | T (°C) |
|---|---|---|---|---|---|---|---|
| East | 2.01 ± 1.10 [c] | 0.10 ± 0.06 [c] | 4.49 ± 2.29 [b] | 7.31 ± 0.76 | 12 ± 6.62 | 3.54 ± 0.35 | 17.66 ± 7.84 |
| Central | 2.28 ± 1.03 [b] | 0.13 ± 0.10 [b] | 4.67 ± 1.51 [b] | 7.46 ± 0.83 | 9.75 ± 5.27 | 3.62 ± 0.39 | 17.47 ± 7.79 |
| West | 4.11 ± 3.87 [a] | 0.27 ± 0.34 [a] | 6.60 ± 6.99 [a] | 7.56 ± 0.90 | 13.04 ± 7.77 | 3.25 ± 0.41 | 17.62 ± 7.41 |

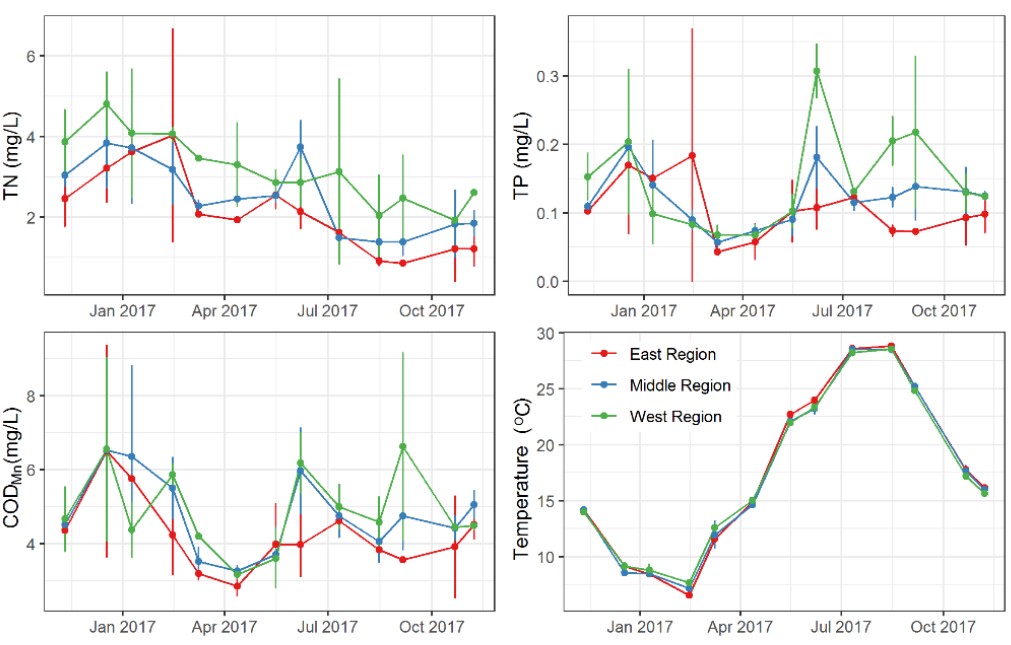

**Figure 2.** Monthly variation in total nitrogen (TN), total phosphorus (TP), permanganate index ($COD_{Mn}$) and water temperature in Lake Chaohu from November 2016 to October 2017.

### 3.2. Seasonal Shift in Phytoplankton Composition

According to the phytoplankton composition results, the Lake Chaohu phytoplankton usually consisted of *Dolichospermum*, *Ankistrodesmus*, *Chlorella*, *Cryptomonas*, *Cyclotella*, *Melosira*, *Microcystis*, *Navicula*, *Pediastrum*, *Scenedesmus* and others (Figure 3). Over the entire lake, *Dolichospermum flos-aquae* and *Microcystis aeruginosa* were the dominant species of phytoplankton and the main contributors to cyanobacterial blooms.

In the eastern area, *Dolichospermum* formed the dominant phytoplankton population, and its biomass was more than 70% for most months. The dominance of *Dolichospermum* changed in April, and the population of *Microcystis* grew rapidly after May. The proportion of *Microcystis* in April, June and October was more than 25%, and it became the dominant phytoplankton population in the eastern area (Figure 3a). In the central area, *Dolichospermum* formed the main phytoplankton population except in July, August and October. *Microcystis* accounted for more than 70% in August and October and became the dominant species (Figure 3b). In the western area, *Dolichospermum* only showed a high proportion in November, December, April, May, and June, and *Microcystis* was the dominant population in January, July, August, September, and October. (Figure 3c)

The environment of Lake Chaohu was in a eutrophic state (Table 1). Compared with the different lake areas, the TP and TN of the western lake were twice as high as those of the eastern and central areas ($p < 0.05$). The $COD_{Mn}$ of the western area was significantly higher than those of the central area and eastern area ($p < 0.05$). There was no significant difference in pH, SD or water depth among the three areas ($p > 0.05$).

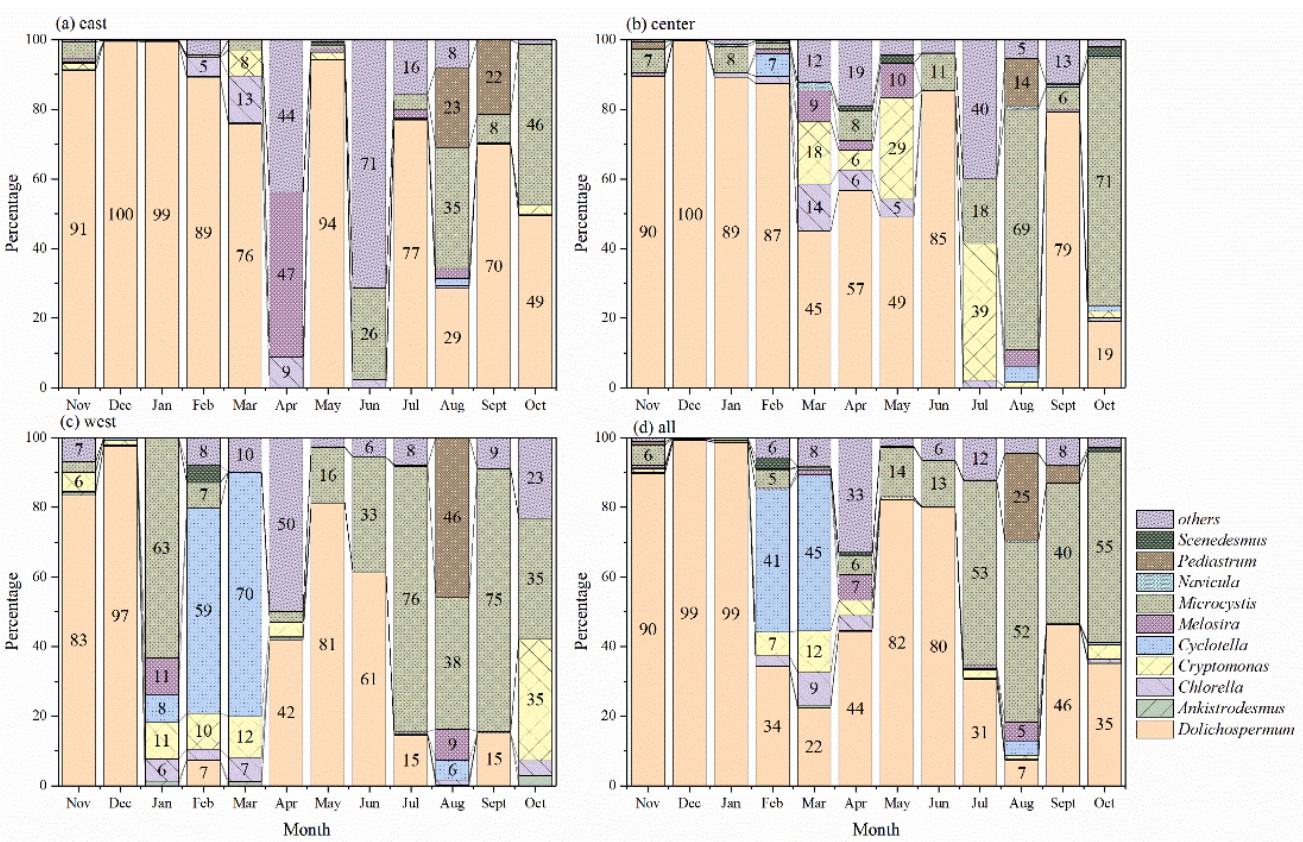

**Figure 3.** Phytoplankton community composition from Nov. 2016 to October 2017 in the eastern area (**a**), central area (**b**), western area (**c**) and whole lake area (**d**) of Lake Chaohu. The numbers in the graph indicate the abundance percentage of these genera.

*3.3. Cell Diameters in Dolichospermum and Microcystis*

During the investigation period, *Dolichospermum* was present during each month in Lake Chaohu. For the lake overall, the average cell diameter of *Dolichospermum* was 7.0 μm in the winter, 4.6 μm in the spring, 3.9 μm in the summer, and 4.8 μm in the autumn seasons. The values of the average cell diameter of the three lake areas were similar to those of the entire lake (Figure 4). The cell diameter of *Dolichospermum* showed an increasing trend from November 2016 to February 2017, increasing from 6.19 μm to 7.3 μm. Then, the cell diameter decreased from 6.58 μm in March to 3.39 μm in April and remained at approximately 4 μm from May to October. The values of cell diameter from November 2016 to February 2017 were significantly higher than those from May to October ($p < 0.05$).

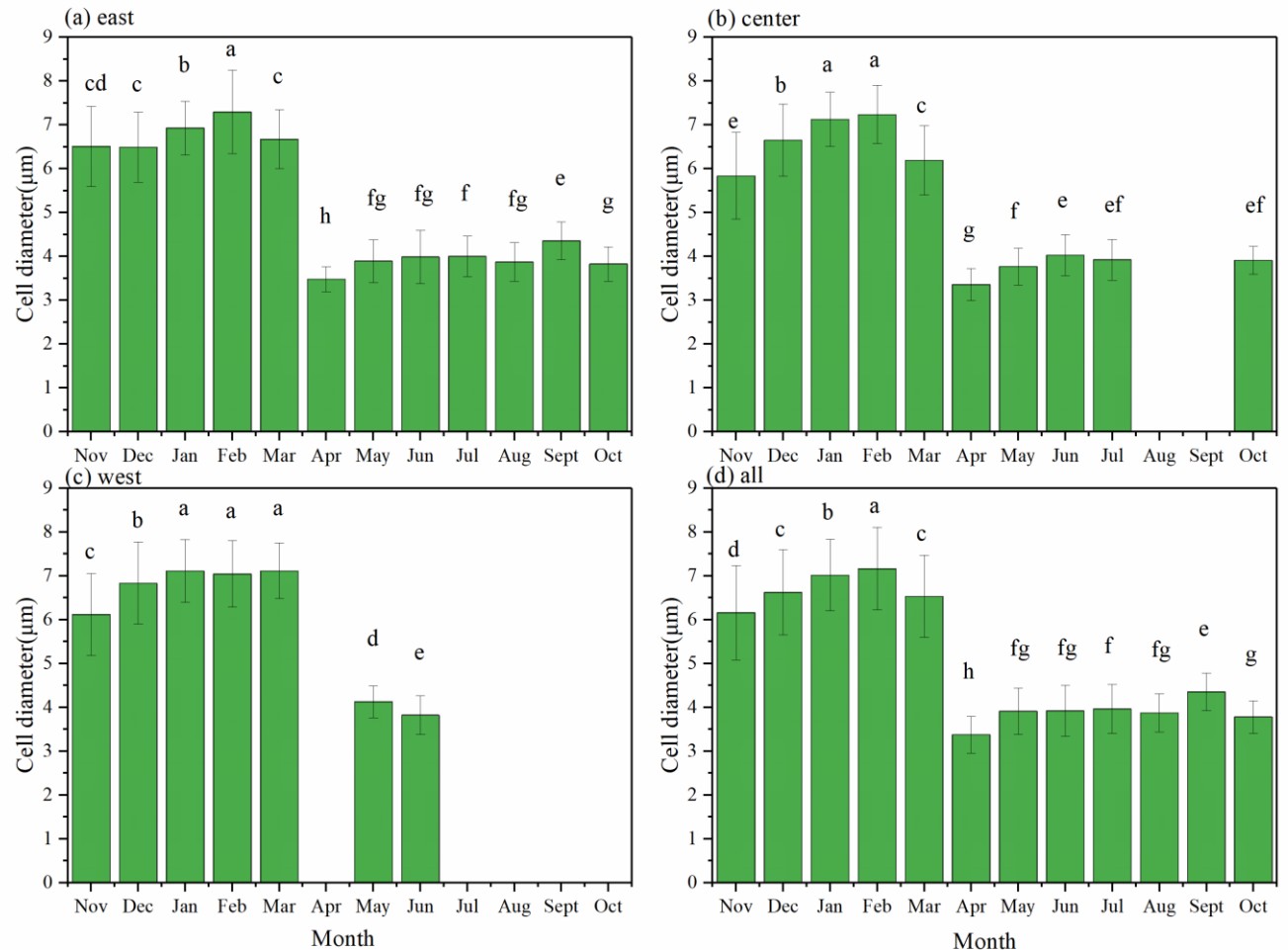

**Figure 4.** The monthly variations in the cell diameter of *Dolichospermum flos-aquae* in the different lake areas. The letters indicate the significance of the differences among the months.

During the investigation period, *Microcystis* was present during most months in Lake Chaohu except December, January, February, March, and April. For the lake overall, the average cell diameter of *Microcystis* was 3.2 μm in the spring, 3.3 μm in the summer, and 4.3 μm in the autumn seasons. No *Microcystis* was present during the winter. The average cell diameters of *Microcystis* were similar, and there was no significant difference among the three lake areas (Figure 5). *Microcystis* generally was not present in December, January, February, or March. The *Microcystis* cell diameter reached its highest value of 5.73 μm in November. The values for cell diameters were significantly lower from April to October than in November.

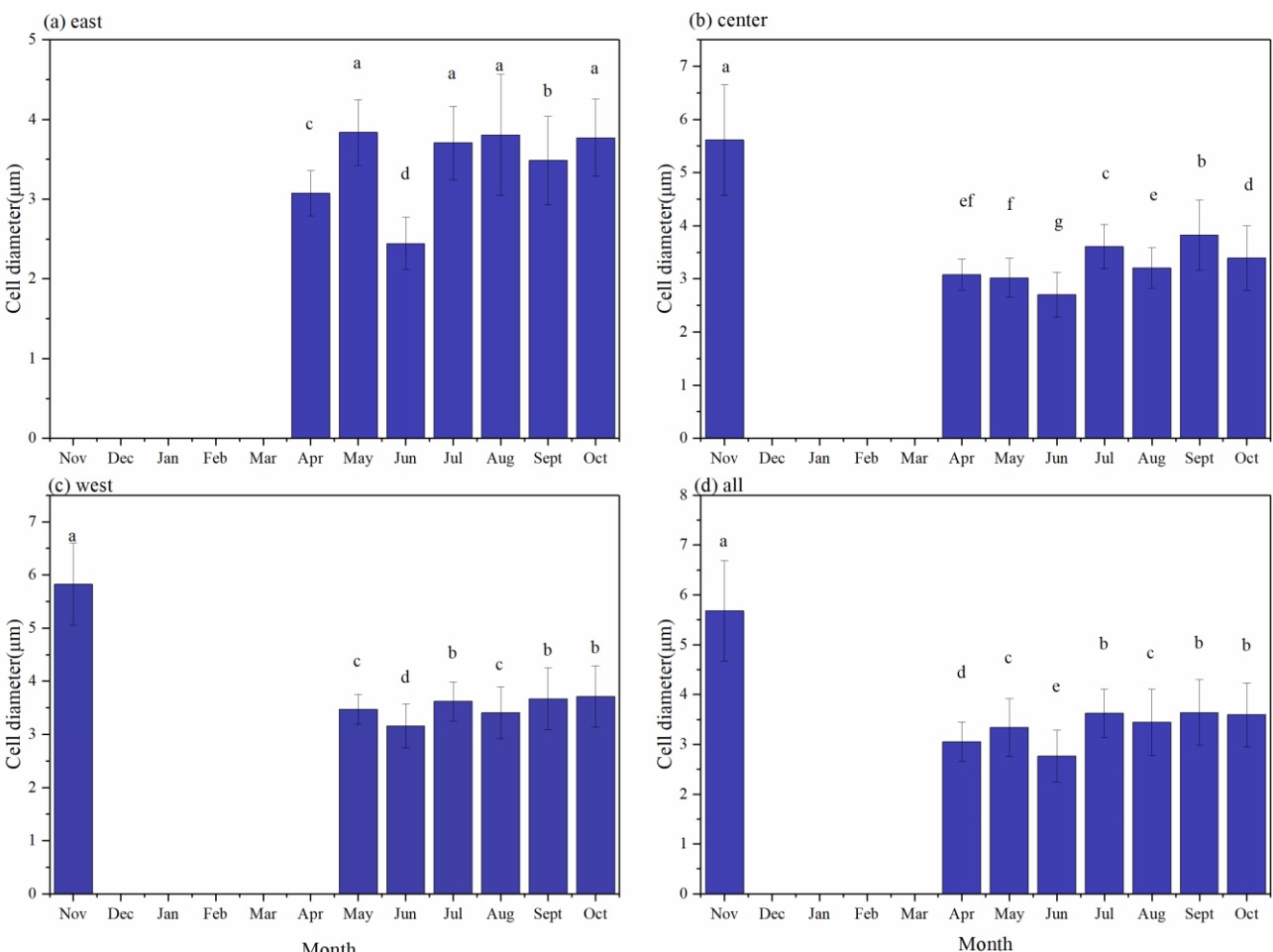

**Figure 5.** The monthly variations in *Microcystis aeruginosa* cell diameter in the different lake areas. The letters indicate the significance of the differences among the months.

*3.4. Colony Size in Different Lake Regions*

According to the frequency histograms of colony size, there were generally two peaks for each lake area and each season. In the eastern area, the colony sizes during the peaks were 174.11 μm and 15.16 μm in September, and the corresponding frequency values were 7.33% and 0.70%, respectively (Figure 6a). By including the results of the phytoplankton composition analysis, we found that *Pediastrum* was the main contributing species for the small peak, and *Dolichospermum* was the main contributing species for the large peak. In December, the colony sizes during the peaks were 8.23 μm and 196.71 μm, and *Dolichospermum* was the dominant species for the large peak. In March, the colony sizes during the peaks were 8.23 μm, 83.71 μm and 409.16 μm. *Dolichospermum* was the main contributing species for the large peak, *Microcystis* contributed to the medium peak, and the small peak was that of *Cyclotella*. In June, the colony sizes during the peaks were 5.71 μm and 174.11 μm; *Microcystis* was the main contributing species for the peak, and other species were the main contributing species for the larger peak.

In the central area, the colony sizes during the peaks were 196.71 μm and 6.45 μm in September. By including the results of the phytoplankton composition analysis, we found that *Dolichospermum* was the dominant species, so both peaks were due to *Dolichospermum*. In December, the colony sizes during the peaks were 196.71 μm and 6.45 μm, and *Dolichospermum* was the main contributor for the two peaks. In March, the colony size during the largest peak was 10.51 μm, and *Dolichospermum* was the main contributing species for the peak. In June, the main peaks were due to *Dolichospermum* and *Microcystis*, and the colony sizes were 196.71 μm and 4.71 μm, respectively (Figure 6b).

In the western area, the colony sizes during the peaks were 6.45 μm and 154.11 μm in September. By including the results of the phytoplankton composition analysis, we found that *Dolichospermum* was the dominant species, and both peaks were due to *Dolichospermum*. In December, the large peak was due to *Dolichospermum*, and the colony sizes were 462.28 μm and 8.23 μm. In March, a peak was observed due to *Cyclotella*, whose colony size was 8.23 μm, and other species. In June, the colony sizes at the peaks were 251.10 μm and 3.1 μm, and *Dolichospermum* and *Microcystis* contributed to the two peaks, respectively (Figure 6c).

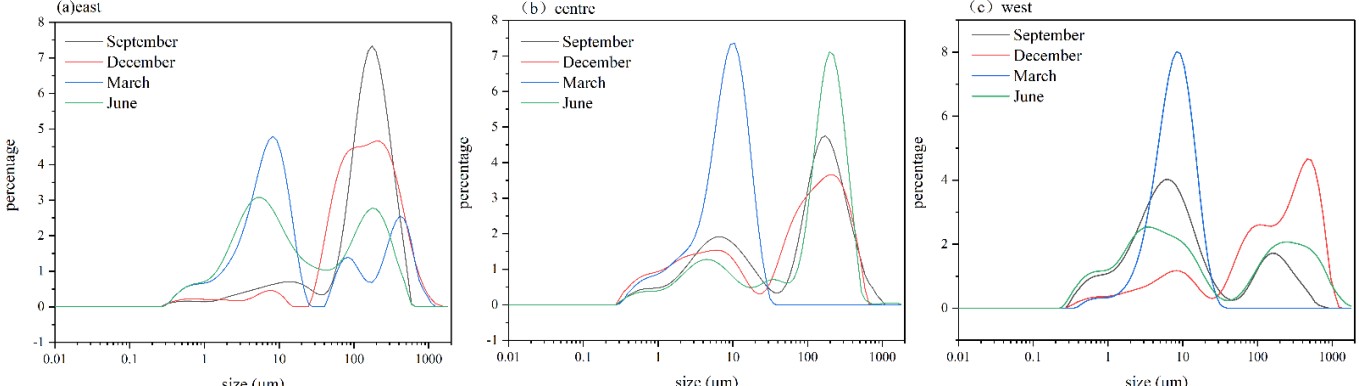

**Figure 6.** Seasonal shifts in colony size in the eastern area (**a**), central area (**b**) and western area (**c**) of Lake Chaohu.

*3.5. Corresponding Relationships between Cell Diameter and Colony Size*

According to the results of linear regression and the generalized linear model, there was a significant relationship between cell diameter and colony size for *Dolichospermum* ($p < 0.05$). The correlation coefficient of the generalized linear model was higher than that of linear regression. There was no significant relationship between cell diameter and colony size for *Microcystis* (Figure 7).

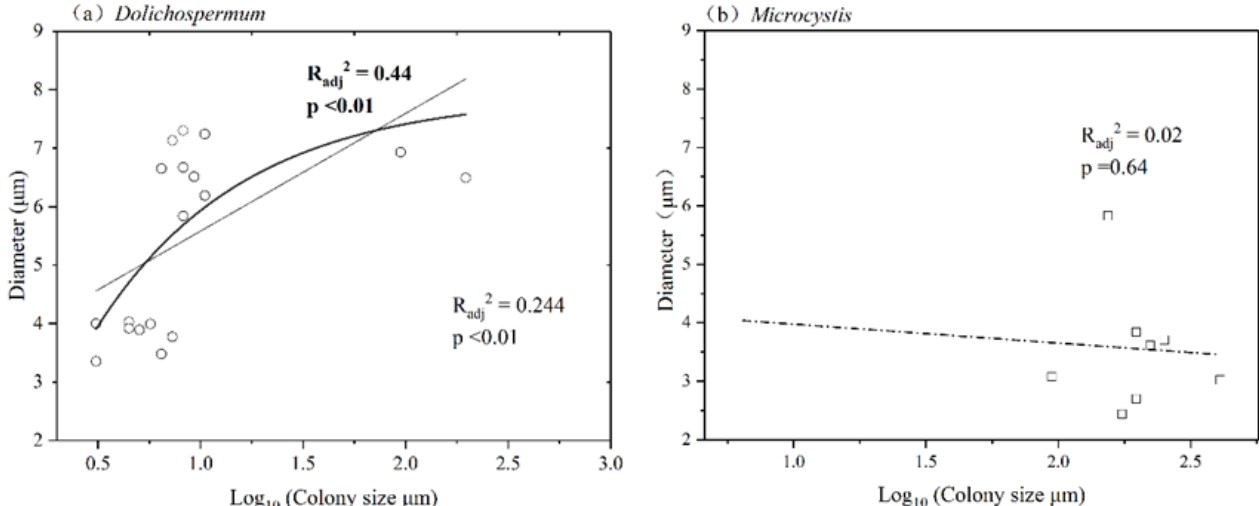

**Figure 7.** The relationship between cell diameter and colony size in *Dolichospermum* ((**a**), the circle) and *Microcystis* ((**b**), the squsre). The solid lines on the left panel were significant fitting lines with linear regression and generalized linear model. The dashed line on the right panel indicated that the linear regression was not significant.

## 4. Discussion

### 4.1. Spatiotemporal Variation in Algal Composition

According to the community composition of phytoplankton, *Dolichospermum* and *Microcystis* were the dominant cyanobacterial genera of phytoplankton in Lake Chaohu. The biomass of *Dolichospermum* and *Microcystis* exceeded 75% of the total biomass during each month. *Dolichospermum* dominated the phytoplankton community in the spring, when the average temperature was lower than 25 °C. *Dolichospermum* blooms mainly occurred simultaneously in the eastern area and central area of Lake Chaohu. During the same time, *Microcystis* was not detected in any of the areas of Lake Chaohu. This phenomenon was consistent with other studies [27]. In the summer and autumn, *Microcystis* grew rapidly with increasing temperatures and became the dominant cyanobacterium. *Microcystis* blooms were concentrated in the western area. After October, the dominant species gradually shifted to *Dolichospermum* with the decrease in temperature. *Microcystis* blooms in the western area gradually decreased, and *Dolichospermum* blooms dominated the eastern area of Lake Chaohu. These findings were consistent with those of our previous study [28], which indicated that there has been no observable change in the community composition of phytoplankton in recent years.

The spatial characteristics of cyanobacterial blooms were directly related to the spatial variation of eutrophication in the lake. The concentrations of nutrients, such those of total phosphorus and total nitrogen, in the western area were higher than those of the central and eastern areas, which was primarily attributed to the input from river runoff in the western region of the catchment [29]. There was a significant positive relationship between TP and *Microcystis* biomass [30,31]. Therefore, it is possible that *Microcystis* was limited by the relatively low TP in the eastern lake area and flourished in the high TP western lake areas. *Dolichospermum* was predominant when nitrogen levels were limited or when the TN/TP ratio was low, because *Dolichospermum* is able to fix $N_2$ when nitrogen is in short supply [32]. However, few heterocysts were observed in the sample dominated by *Dolichospermum*, suggesting that nitrogen availability may have been sufficiently high, and that nitrogen fixation by heterocysts was not unnecessary for *Dolichospermum* dominance. Therefore, the variations in the TN/TP ratio or in the low TN concentration regions may not have been the primary factor driving the dominance of *Dolichospermum* in Lake Chaohu. In this study, *Dolichospermum* primarily dominated in the spring, autumn and winter, when the temperature was generally relatively low, which indicates that they might have a competitive advantage in the low-temperature niche.

### 4.2. The Variation in Colony Size and Cell Diameter

In this study, *Dolichospermum* cell diameter increased from 6.5 µm to 7.4 µm during the period from November 2016 to March 2017 and then decreased significantly from approximately 7 µm during the period from November 2016 to February 2017 to approximately 4 µm during the period from May to October. The *Microcystis* cell diameter values were also significantly lower from April to October than in November. The increase in cell diameter during a period of relatively low temperatures may be attributed to a growth strategy in which cyanobacteria store energy to survive low temperature conditions, which is confirmed by the finding that spermidine promotes gene expression in cyanobacteria and promotes the absorption of nutrients to improve cyanobacterial ability to overwinter [33].

The cell diameters of the two cyanobacteria decreased primarily during the growing season. Before the growing season, the two cyanobacteria experienced a few months of overwintering, which depleted the energy stores of the cyanobacteria [34]. Therefore, as the cyanobacteria began to enter their breeding process after dormancy, their cell diameters detected in April were significantly smaller than before. In addition, as the cyanobacteria began to growth, the energy absorbed from nutrients was mainly used for their growth process rather than being stored in the cell for rapid reproduction. This regulation in nutrient utilization strategies might contributed to the variation of cell diameters.

The colony sizes in Lake Chaohu showed distinct seasonal variations. The cyanobacteria colony size decreased from September 2016 to March 2017 and then increased from March to June. The increases in the colony sizes of the two bloom-forming cyanobacteria were helpful for improving their floatation ability and for reducing predation by zooplankton. Buoyancy facilitates light acquisition and reduces predation [35]. The vertical distribution of the cyanobacterial species reflects the autecological preferences of the cyanobacteria for the prevailing environmental conditions and also reflects the minimization of negative interactions and exploitation of positive interactions [36]. The decrease in colony size might be due to sinking after flourishing or due to the shedding of individual cells from the colony.

The formation and growth of colonies was considered to be due to the incomplete separation and due to adhesion of daughter cells after cell division [37]. When a colony grows to a certain size, some cells separate and become propagules for new colony growth. This is also consistent with our previous studies showing that single-celled *Microcystis* form colonies under predation pressures from zooplankton, and the formation and growth of these colonies is also the result of incomplete separation after cell division [38]. Environmental factors, such as light, nutrients and temperature, have important effects on the formation and size of cyanobacterial colonies [39]. The colony size of *Microcystis* is known to decrease with increasing temperature or nutrient concentrations despite faster growth of unicellular cells [40,41]. Higher light intensities led to a faster growth rate of cells, which accelerates the consumption of intracellular polysaccharides and other substances, and decrease the propensity to form colonies [42]. In the study, however, no significant differences in the cell diameters or colony size of *Microcystis* and *Dolichospermum* were observed among the lake regions with obvious different nutrient levels. This is not consistent with the previous study, perhaps due to the gradient of nutrients or combined effect of multiple variables. As such, this aspect requires further study in the future.

In this study, we found that there were positive relationships between the colony size and cell diameter of *Dolichospermum*, while there were no such significant relationships for *Microcystis*. During cyanobacterial blooms, the biomasses of *Microcystis* and *Dolichospermum* were highly correlated with cell diameter. These results indicate that *Dolichospermum* and *Microcystis* may absorb more nutrients by adjusting colony size and cell diameter, thus gaining competitive advantages. With the increase in *Dolichospermum* biomass, the available nutrient concentrations decreased, which decreased access to nutrients. We speculate that *Dolichospermum* changes its growth strategy, growing rapidly while maintaining low cell diameters to increase cell content and colony size. This strategy could facilitate the maintenance of a higher biomass level; therefore, *Dolichospermum* and *Microcystis* might maintain their biomass by inducing a trade-off between cell diameter and colony size.

## 5. Conclusions

In summary, *Dolichospermum* and *Microcystis* were found to be the dominant species of the cyanobacterial blooms in Lake Chaohu. Colony size and cell diameter in *Dolichospermum* were significantly and positively correlated, while no such relationship was observed for *Microcystis*. The regulation of cell diameter and colony size in *Dolichospermum* might be a mechanism by which it gains a competitive advantage and maintains biomass. Differences in cell diameter and colony size in response to variations in seasonal environment changes may be due to the unique physiological advantages that *Microcystis* and *Dolichospermum* developed during their respective evolutions. Our study highlights the importance of morphological regulation in maintaining bloom-forming cyanobacterial biomass. However, our understanding of the mechanism of the trade-offs among morphological traits is still limited. It is necessary to further investigate the process and factors affecting these trade-offs.

**Author Contributions:** Conceptualization, M.Z. and X.S.; validation, Y.M. and M.Z.; formal analysis, Y.M.; investigation, Z.Y., L.S. and Y.Y.; writing—original draft preparation, Y.M.; writing—review and editing, M.Z.; funding acquisition, M.Z., X.S. and Z.Y. All authors have read and agreed to the published version of the manuscript.

**Funding:** This work was supported by grants from the National Natural Science Foundation of China (32171546, 31870447, 32071573, 31971476), the Key Research Program of Frontier Sciences, CAS (Grant No. ZDBS-LY-DQC018), Science and Technology Service Network Initiative of Chinese Academy of Sciences (KFJ-STS-QYZD-2021-01-002) and Science and Technology Achievement Transformation Foundation of Inner Mongolia Autonomous Region (2021CG0013).

**Institutional Review Board Statement:** Not applicable.

**Informed Consent Statement:** Not applicable.

**Data Availability Statement:** The datasets generated during and/or analyzed during the current study are available from the corresponding author on reasonable request.

**Acknowledgments:** We thank Kuimei Qian for the identification and counting of phytoplankton samples. We also greatly thank anonymous reviewers for reviewing the manuscript.

**Conflicts of Interest:** The authors declare no conflict of interest.

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
