# Peer review of "Seasonal Shifts in the Morphological Traits of Bloom-Forming Cyanobacteria in Lake Chaohu, China"

_diversity, doi:10.3390/d14060435_

Round 1

Reviewer 1 Report

The manuscript (diversity-1738129) "Seasonal shifts in the morphological traits of bloom-forming cyanobacteria in Lake Chaohu, China” submitted by Meng et al. performed a one-year-long phytoplankton survey in Lake Chaohu, China, and measured the colony size and cell diameter of the dominant cyanobacteria, Microcystis and Dolichospermum. They found that Dolichospermum may maintain biomass through a trade-off between cell diameter and colony size and that there is a flexible morphological regulatory mechanism. This study can help improve our understanding of how bloom-forming cyanobacteria maintain their dominance by regulating their morphological traits.

The following are my major and specific comments:

  1. As we all know, there is an obvious shift of species composition, even within one genus. For example, throughout a year, Microcystis might shift from Microcystis flos-aquae to Microcystis aeruginosa, to Microcystis wesenbergii. Were all the species or only Microcystis aeruginosa were measured?
  2. How did you guarantee that the colonies belong to one species during the process for measuring the colony size with a laser particle size analyzer?
  3. L35: affects
  4. L110: Please give the full name of the CODMn.
  5. L149: the name of the dominant cyanobacteria should be shown in species level
  6. L165: replace the Secchi with SD, and check that throughout the manuscript
  7. L230: Reorganize the sentence “There was no significant relationship for Microcystis”.
  8. L285-287 and L290-292: These two sentences are inappropriate to explain why the two cyanobacteria have smaller diameter during the growing season.
  9. L301-302: soften the tone of the sentence
  10. L323-326: delete the sentence.

Figures:

  1. 3 & 4. Indicate the significant difference with letters from HSD test results.
  2. 2, 3 &4. Change the number in x-axis label to the Abb. for months in English terms.
  3. 5. The percentage reference of the ordinate in is not clear.
  4. The resolution of all figures needs to be improved.

Author Response

Dear Reviewer,

Many thanks for your valuable comments and suggestions, which are very important to improve our manuscript. We have modified the manuscript according to the comments, and responded to the comments one by one as following. We sincerely hope that these modifications can make the revision logical and easy to be understood for you and more readers, and reach the standard for publication in the journal.

Thank you again for your reviewing work.

Best wishes

Min Zhang

Reviewer 2 Report

Review for the paper "Seasonal shifts in the morphological traits of bloom-forming cyanobacteria in Lake Chaohu, China" by Yangyang Meng, Min Zhang, Zhen Yang, Xiaoli Shi, Yang Yu, Limei Shi submitted to "Diversity".

General comment.

The authors studied seasonal dynamics of functional traits in different bloom-forming cyanobacteria inhabiting Lake Chaohu, China. Phytoplankton assemblages play a key role in the aquatic systems and form primary production in any water body. Therefore, the paper may be interesting for many ecologists focusing on succession in freshwater ecosystems. The present study showed clear seasonal changes in composition and size structure of common cyanobacteria. Some species were found to be able to maintain biomass through a trade-off between cell diameter and colony size. The authors highlighted the importance of cyanobacteria morphology to support their prevalence in the freshwater phytoplankton. A comprehensive data set was used to obtain the results. Methods to collect and process the plankton samples seem to be adequate. Statistics and data treatment are relevant and allow obtaining valid results. Main findings are well-illustrated and discussed in a clear manner. However, I think that the main results should be interpreted more thoroughly. I also have some minor recommendations to improve the ms.

Major concern.

Discussion is short and may be improved by adding more text explaining ecological mechanisms determining variations in the cell diameter and colony size in the common cyanobacteria.

Also, please explain more carefully which ecological changes may be related to observed dynamics in cell size and morphology of the microalgae. Which trophic levels may be altered due to these seasonal fluctuations?

Specific remarks.

L57. 'Alexandrina' should be italicized.

L84. Consider replacing "adjust" with "regulate".

L107. Please, provide total number of phytoplankton samples obtained in the study.

L113. Consider replacing "aspirated" with "removed".

Table 1 caption. Please, update the caption with definitions of the abbreviations TN, TP, COD, pH, and SD.

Table 1 must be updated with the water temperatures in each lake region.

Also, I encourage the authors to provide a combined graph showing seasonal pattern of the environmental data.

L115. Please, provide magnification.

L161-165. I suggest to move environmental background in the beginning of the section 3.1.

I would like also to see a short description regarding the seasonal dynamics of environmental variables during the study period.

L168. Consider replacing "were" with "indicate".

Figure 3, 4. What do mean vertical bars?

L323-325. Non -relevant text.

Author Response

(The authors gave the same response as above.)

Round 2

Reviewer 2 Report

No additional comments.